# Biologic Brachytherapy: Genetically Modified Surgical Flap as a Therapeutic Tool—A Systematic Review of Animal Studies

**DOI:** 10.3390/ijms251910330

**Published:** 2024-09-25

**Authors:** Wiktor Pascal, Mateusz Gotowiec, Antoni Smoliński, Michał Suchecki, Michał Kopka, Adriana M. Pascal, Paweł K. Włodarski

**Affiliations:** 1Department of Methodology, Medical University of Warsaw, 1b Banacha Street, 02-091 Warsaw, Poland; mateusz.gotowiec@wum.edu.pl (M.G.); s082979@student.wum.edu.pl (A.S.); s082981@student.wum.edu.pl (M.S.); michal.kopka@wum.edu.pl (M.K.); adriana.paskal@gmail.com (A.M.P.); pawel.wlodarski@wum.edu.pl (P.K.W.); 2Doctoral School, Medical University of Warsaw, 81 Żwirki i Wigury Street, 02-091 Warsaw, Poland

**Keywords:** surgical flap, gene therapy, animal study, translational research, biologic brachytherapy

## Abstract

Surgical flaps are rudimentary tools in reconstructive surgery, especially following extensive solid tumour resections. They cover skin and soft tissue defects but are prone to ischaemia and necrosis. Since their primary aim is reconstruction, they rarely exhibit a therapeutic activity against the treated disease. Attempts have been made to develop a new therapeutic strategy—biologic brachytherapy, which uses genetically engineered surgical flaps as a drug delivery vehicle, allowing the flap tissue to act as a “biologic pump”. This systematic review summarizes the preclinical evidence on using genetically modified surgical flaps. A literature search was conducted in PubMed, EMBASE, Scopus and Web of Science. The initial literature search yielded 714 papers, and, eventually, seventy-seven studies were included in qualitative analysis. The results show that genetic enhancement of flaps has been used as a local or systemic therapy for numerous disease models. Frequently, it has been used to increase flap survival and limit ischaemia or promote flap survival in a non-ischemic context, with some studies focusing on optimizing the technique of such gene therapy. The results show that genetically modified flaps can be successfully used in a variety of contexts, but we need more studies to implement this research into specific clinical scenarios.

## 1. Introduction

For the last two decades, major developments have occurred in the reconstructive surgery field. The progress from random-pattern skin flaps with limited vascular support towards more complicated composite and hybrid flaps with a microvascular connection, increases overall survival and limits potential complications such as ischaemia, venous congestion or lymphatic drainage dysfunction. However, distal flap necrosis is still a major complication of all flap surgeries [1]. Numerous techniques have been used to increase flap survival including pre-operative, intra- or post-operative approaches [2,3]. However, most of the techniques have short-term effects and require repetitive administration. As a response, biological-based therapies are being developed, recently focusing on the use of stem cells [4], but effective and feasible interventions are still lacking. Another approach is using further enhanced techniques, including genetically modified vectors delivered directly to the flap, as they allow more long-term effects and limited systemic toxicity. For the first time, such a technique was successfully used in 1996 by Takeshita et al. [5], paving a way for a new method.

Furthermore, the early developments of novel gene therapy-based techniques focusing on increasing flap survival were followed by the idea that a flap may not only serve the purpose of helping wound healing and recreating tissue architecture but also become a therapeutic source itself. This concept was first proposed by Michaels et al. [6,7] in 2004 and named “biologic brachytherapy”, defined as the genetic modification of tissue bulk for its use as a local or systemic therapeutic. Thanks to this innovative technique, the surgical flap may function as a reconstructive and therapeutic tool at the same time (Figure 1). Since then, major developments have been made in the field.

This systematic review aimed to appraise the evidence on the use of genetically modified surgical flaps and establish the main areas of research with the largest developments.

## 2. Materials and Methods

This study is an extension of the meta-analysis protocol, registered with PROSPERO (CRD42022359982). Data gathered in the previous study have been re-analysed, and more studies have been included. This study followed the PRISMA (Preferred Reporting Items for Systematic Reviews and Meta-Analyses) guidelines for reporting systematic reviews [8]. Here, we only summarized the study protocol, which is available in full as a Appendix A and in the PROSPERO database using the aforementioned registration number.

### 2.1. Study Selection

All preclinical studies that used a genetic modification of a surgical flap as a therapeutic or study tool were included.

### 2.2. Search

The following primary MeSH terms were used: Bulk: Genetic therapy [mh] OR “Gene* therapy” OR “Gene* insert*” OR “Gene* modif*” OR “Modif* gene*”///“gene* transfect*” Cellular component: Cell transplantation [mh] OR “cell* transplant*” OR “MSCs” OR “mesenchymal stem cell*” OR mesenchymal stem cells [mh] Viral vectors: (“adenovir*” OR “Retrovir*” OR “herpes simplex vir*” OR “vaccinia vir*” OR “lenti-vir*”) AND “vector*” Bacteria: “bacter*” OR “plasmid*” OR bacteria [mh] Other: “electroporat*” OR “ultraso*” OR “ballistic delivery” OR “particle* bombardment” OR “gene gun” OR “liposom*” OR drug carriers [mh] OR “plasmid*” OR “receptor-mediate*” OR “polymer*”. The specific search strategy is available in the PROSPERO protocol. The last date of search was 30 November 2022. Updated on 7 July 2024.

### 2.3. Data Extraction

Titles and abstracts of studies retrieved using the search strategy were screened using Rayyan QCRI to identify studies that potentially met the inclusion criteria. All screening was carried out independently by two review team members (M.G. and M.K.). Any disagreement between them over the eligibility of particular studies was resolved through discussion with a third author (W.P.) where necessary.

## 3. Results

The initial literature search yielded 714 papers; the number was reduced to 626 after duplicate removal. After abstract screening in Rayyan, 79 papers were eligible for full-text retrieval, 69 of which fulfilled the inclusion criteria. Eight more eligible records were identified via hand-searching. Eventually, 77 articles were included in the qualitative analysis (Figure 2).

All included studies were divided into four groups:Interventions focusing on affecting surrounding tissue or the whole organism using the flap and achieving additional therapeutic activity;Preconditioning interventions directed towards affecting surroundings and the flap;Interventions focusing on improving flap survival and managing flap surgery-related complications;Studies that described the safety and efficacy of surgical flap gene therapy.

### 3.1. Local or Systemic Tissue Therapy Using Genetically Modified Surgical Flaps

We found eleven studies that aimed at using genetic flap modification to achieve additional therapeutic activity [7,9,10,11,12,13,14,15,16,17,18] (Table 1). In total, those studies were performed with at least 309 animals, as one study did not report the number of animals. We have subdivided the studies into four distinct groups depending on the area of research.

The first subset contained attempts to improve wound healing. Casal et al. [9] focused on inflammation as an aspect of chronic wounds. In a Pseudomonas aeruginosa infection model, the flap was modified with human beta-defensins (BD-2 and BD-3). β-defensins are cysteine-rich peptides that physiologically take part in the innate immune response. They act as proinflammatory mediators and chemo-attractants for macrophages, T cells and dendritic cells but are also able to directly kill microorganisms, potentially through the disruption of cell wall synthesis [19]. The best flap survival was measured for the group with BD-3, with a statistically significant increase of living tissue compared to the control. Another antimicrobial agent was proposed by Ghali et al. [10], where the human cathelicidin (LL-37) gene was used in modified viruses to decrease the number of bacteria around an implanted catheter. Human cathelicidin (LL-37) is the only representant of cathelicidin present in humans, which, along with defensins, constitute antimicrobial peptides (AMPs). LL-37 has moderate antimicrobial activity, both against Gram-negative and Gram-positive bacteria and, similar to defensins, has immunomodulatory potential [20]. Results showed a significant increase of bactericidal activity of such treatment. Saad et al. [11] used a different strategy, improving wound healing within an ischemic skin flap by enhancing angiogenesis. Thioredoxin-1 was used as a stimulator for vessel growth. Thioredoxin-1 is a cytosolic multifunctional protein, first described as a hydrogen donor for enzymes [21], yet currently known to be an important molecule in protection from ischaemia reperfusion, where it acts as a redox regulator of reactive oxygen species and transmits anti-apoptotic signals [22]. In the ischemic wound model, the percentage of closed wounds after nine days was almost twice as high in the experimental group compared to the untreated control.

Except skin wounds, bone healing improvement was also studied. Lampert et al. [12] modified a quadriceps femoris muscle flap with bone morphogenetic protein 2 (BMP-2). The bone morphogenesis proteins (BMPs) are growth factors that promote the formation of bone tissue, especially in an early response to trauma or injury. Among them, BMP-2 is considered the strongest inductor of growth, with not only osteogenic ability but also the ability to cause bone marrow mesenchymal stem cells to differentiate into osteoblasts [23]. The regeneration of the femur after an iatrogenic defect was observed for 14 weeks. As a result, animals from the experimental group had significantly higher density of bone compared to the control. Aliyev et al. [13] concentrated on creating a new modality for osteomyelitis treatment. In their study, researchers applied plasmids with VEGF-165 to muscle flaps and transposed them over to the inflamed region. The results were compared to a nontreated muscle transposition group and a group that received only antibiotics. The results showed no significant gain from plasmid treatment; however, both treatments were superior to the antibiotics-only group, showing limited efficacy in using VEGF enhancement of a surgical flap in this context.

A number of studies concerned anticancer therapies. Dempsey et al. [14] proved the efficacy of flap transfection with IL-12 via an adenoviral vector in a breast cancer model. Interleukin-12 is a pro-inflammatory cytokine produced typically by antigen presenting cells in response to a bacterial infection, however it also acts as a potent inducer of cell-mediated immunity. However, its use is limited by excessive side effects and toxicity after systemic administration; therefore, localized delivery methods are sought [24]. Average tumour size by the end of the study was significantly lower in the experimental group, and the concentration of β-hCG produced by cancer cells was also significantly lower in the treatment group. Seth et al. [15] used an adenovirally modified epigastric flap to produce a virally delivered prodrug (VDEPT) consisting of herpes simplex virus-thymidine kinase (HSV-TK)/ganciclovir (GCV) to treat a rat model of residual disease using glioma cells. HSV-TK is a suicide gene that, when introduced into the cell, phosphorylates the prodrug GCV to its active triphosphorylated form, which can inhibit DNA synthesis and induce apoptosis [25]. Flaps were placed near micro- or macro-residual tumour models. Results showed significantly extended time to palpable tumour mass detection and increased median survival in the experimental group. Another study used a similar model to treat breast cancer in rats. Zhang et al. [17] used a lentivirus to deliver two double suicide genes, *Escherichia coli*-derived cytosine deaminase (CD) and thymidine kinase (HSV-TK), together with corresponding drugs, GCV for HSV-TK and 5-flucytosine, which is deaminated by CD to 5-fluorouracile, a potent RNA and DNA synthesis inhibitor [26]. This therapy also reduced the volume and mass of tumours compared to the control; however, it is difficult to assess its superiority compared to using a single VDEPT therapy when delivered using surgical flaps. The last study, by Davis et al. [18], used a different approach by using adenovirally transduced stromal vascular fraction (SVF) cells, consisting primarily of mesenchymal progenitor/stem cells, to produce interferon-gamma (IFNγ) locally in a rat model of breast cancer. Interferon gamma is a pleiotropic cytokine, with antiviral, immunomodulatory and antitumor properties, physiologically secreted by CD4 helper and CD8 cytotoxic T cells, activating local M1 macrophages [27]. In this study, IFNγ secreted by transduced autologous SVF cells was shown to activate M1 macrophages for four weeks. As fluorescently labelled cancer cells were used in this model, a significantly lower cancer emission signal was present in the treatment group, as well as a significantly longer overall survival.

The last subset of studies considered transcending the idea of local tissue therapy. Michaelis et al. [7] proposed this idea for the first time and tested the possibility of delivering endostatin, a small cleavage fragment from the α1 chain of type XVIII collagen that has antiangiogenic and anticancer potential, using adenovirally transduced free flaps. Their results showed a significantly high concentration of endostatin in serum for up to 10 days (last reported time-point) since the flap elevation. However, as this study focused on the possibility of the systemic delivery of anticancer therapeutics and no cancer model was used, it is difficult to determine its treatment efficacy. Another study that tested such an approach was undertaken by Than et al. [16]. They used minicircle DNA containing human coagulation factor IX, which is missing in patients with hemophilia type B, transfected the entire free flap with intraarterial perfusion and reimplanted the flap back into the same site. Thanks to this therapy, a significant systemic expression of coagulation factor IX was detected for much longer—reported for the last time at four weeks, with no further follow-ups—which shows that genetically modified surgical flaps may serve not only as a local disease treatment but potentially as a long-term therapeutic tool for systemic protein deficiency disorders.

### 3.2. Genetic Flap Preconditioning Affecting Surrounding Tissue and the Flap

Another group of six studies focused on achieving both additional therapeutic activity and acting as flap-directed therapy [28,29,30,31,32,33] (Table 2). The aim was flap preconditioning using genetic vectors to later promote flap survival; however, the modelled diseases transgressed typical flap-surgery complications, and the flaps actively acted as source of additional biologic activity affecting surrounding tissues. Two studies aimed to prevent post-radiation flap necrosis, while another four intended to modulate the immune response in the allotransplantation of adipocutaneous flaps.

In two studies regarding the management of irradiation damage, a total of eighty-five rats were used [28,29]. Angelos et al. [29] conducted a study in which a plasmid enriched with the vascular endothelial growth factor (VEGF) gene was applied to irradiated ventral fasciocutaneous flaps. When ligation of the pedicle was performed fourteen days after plasmid DNA application, flap revascularization was significantly improved. More complex research was conducted by Khan et al. [28], where, in the first part of the experiment, superoxide dismutase 2 (SOD2) was upregulated, and connective tissue growth factor (CTGF) was inhibited in a superficial inferior epigastric artery (SIEA) flap using lentiviral vectors. SOD2 is a metalloenzyme located in the mitochondrial matrix, involved in the catalysis of ROS, which are typically associated with radiotherapy (RT) [34]. CTGF is the main mediator of fibrosis and tissue remodelling, which promotes the development of late-adverse effects following RT [35]. In the following step, animals were injected with breast carcinoma cells and then irradiated for five days. Both interventions resulted in a decrease in the relative surface of skin paddle contracture and less skin volume loss compared to the control, showing the effectiveness of such therapy in skin damage following RT.

Four studies with at least 112 rats focused on enhancing adipocutaneous flap allotransplantations [30,31,32,33]. A study conducted by Fu et al. [30] revealed that the application of an adenoviral vector for OX40 immunoglobulin (OX40 Ig) combined with rapamycin provided a significantly longer time of flap survival after orthotropic SIEA flap transplantation. The CD134 molecule, also called OX40, is a costimulatory molecule associated with T-cell activation and proliferation upon activation by the OX40 ligand (OX40L), present on activated antigen-presenting cells [36]. OX40 Ig was used to block this pathway through binding to OX40L, inducing longer allotransplant survival and limiting the inflammatory response. Research conducted by Xiao et al. [31] focused on another immunosuppressive particle, cytotoxic T-lymphocyte-associated protein 4 (CTLA4), which, by binding to B7-1 (CD80) or B7-2 (CD86), can switch off antigen-presenting cells, leading to a major disruption of the cell-mediated immune response [37]. In this study, flaps were perfused with an adenoviral vector containing a gene for CLTA4 immunoglobulin (CTLA4 Ig). Significantly better results were achieved after treatment with CLTA4 Ig and rapamycin compared to the rapamycin-only group. Another study concerning allotransplantations was conducted by Zhang et al. [32] and combined both aforementioned immunoregulatory proteins. The experimental groups were treated with lentiviruses encoding CTLA4 Ig and OX40 Ig independently and in combination, as well as additional rapamycin. The group treated with both CTLA4 Ig and OX40 Ig had the longest time of survival, especially compared to the rapamycin-only group. Nonetheless, only qualitative data were available on flap survival, showing that on the 28th day, the control group noted flaps necrosis, whilst any kind of genetic modification prevented it. The last study in this group, undertaken by Jeong et al. [33], focused on using a different molecule, adenovirally delivered rat intereleukin-10, previously described as an important immunomodulator. Viral particles were injected intravascularly into an epigastric flap transplanted from Sprague-Dawley to Wistar rats. The results showed a significantly reduced acute rejection response; however, the graft survival in days was less than two, showing the limitations of such therapy.

### 3.3. Genetically Modifie Surgical Flap for Flap-Surgery Related Complications Management

Although commonly used in plastic and reconstructive surgery, surgical flaps, both free and pedicled, can suffer from ischaemia followed by the necrosis of the transplanted tissues. A variety of factors are considered as mediators and effectors of the stress response following ischaemia/reperfusion injury, including proinflammatory signals such as the NF-kB pathway, TNF-α, IL-1β and interleukin-6 [38], electrolyte dysregulation—primarily an increase in potassium and sodium ion concentration in extracellular fluid due to cell membrane rupture and intracellular fluid leakage [39]—as well as an accumulation of reactive oxygen species (ROS), followed by the recruitment of immune cells [38]. Due to the plethora of potential targets, the main aim of the genetic modifications of flaps is to promote angiogenesis and vascularization, limit the inflammatory response and modify extracellular matrix properties (Figure 3).

Among the screened papers, 58 focused on increasing ischemic flap survival by diverse means—from directly delivering transgene containing plasmids to transplanting genetically modified cells for local or systemic expression [40,41,42,43,44,45,46,47,48,49,50,51,52,53,54,55,56,57,58,59,60,61,62,63,64,65,66,67,68,69,70,71,72,73,74,75,76,77,78,79,80,81,82,83,84,85,86,87,88,89,90,91,92,93,94,95,96,97] (Table 3).

Our previous meta-analysis provided thorough insights into the efficacy of such therapies in promoting flap survival [98]. Therefore, here, we only summarized the report. In total, 2853 rats of several strains, 32 rabbits and 221 mice were included in the studies. The most common primary endpoint was flap survival at follow-up, ranging from seven to fourteen days, with seven days post-flap elevation results being the most common. The most commonly described target proteins and pathways used to promote vascularisation included the following:
Vascular endothelial growth factor (VEGF), a signalling molecule that belongs to the PDGF supergene family and, as a homodimer, regulates angiogenesis and vascular permeability (VEGF-A) [99].Platelet-derived growth factor (PDGF), a heterodimer cytokine, which promotes angiogenesis [100].Fibroblast growth factors (FGFs), with leading member FGF-2, which belong to heparin-binding growth factors, contributing to angiogenesis via regulation of the SRSF1/SRSF3/SRPK1 network, leading to VEGFR-1 alternative splicing [101].Hepatocyte growth factor (HGF), which acts through the Met receptor and induces the expression of VEGF leading to an angiogenic response [102].Angiopoietin 1 (Ang-1), a cytokine required for vessel maturation, which mediates the migration, adhesion and survival of endothelial cells [103].Hypoxia-inducible factor 1α (HIF-1α), an important transcription factor regulating the transcription of genes associated with angiogenesis, primarily VEGF [38].Interleukin 10 (IL-10), an anti-inflammatory interleukin, with limited understanding regarding its influence on vessel development with potential pro- and anti-angiogenic actions [104].Transforming growth factor beta (TGF-β), a pleiotropic factor that participates both in vasculogenesis and angiogenesis, resulting in the promotion or suppression of endothelial cell migration and proliferation [105].

### 3.4. Genetically Modified Flaps Used for Optimization Studies

Finally, we analysed two studies that focused only on optimizing technical aspects of performing genetic modification of the flaps [6,106] (Table 4).

Michaels et al. [6] tested different conditions of flap perfusion with adenoviruses with the β-galactosidase gene. The systemic injection into the tail vein group demonstrated a broad, low-level β-galactosidase activity in all harvested tissues, whereas in the ex vivo flap injection group, β-galactosidase activity was 20-fold higher in the transduced flap than in any other tissue. Additionally, both intramuscular injection and intravascular flap perfusions yielded comparable localized β-galactosidase activity. Agrawal et al. [106] focused on optimizing the adenoviral delivery of luciferase gene to the SIEA flaps. The highest luciferase activity was observed in intravascularly injected flaps, medium activity was seen in non-specific injections to flap tissues and the weakest detection concerned topical bathing with no gene expression in the internal organs of the animals (spleen, liver, heart, testes, kidneys).

## 4. Discussion

The gathered results show a wide range of potential applications of genetically modified surgical flaps, ranging from increasing flap vascularization and therefore survival, to using it as an antitumor therapy modality (Figure 4). In the included studies, more than 3000 animals were used to verify the above. Such a number of animals suggests a thorough analysis for the preclinical phase. However, to date, these results have not been translated into the clinical approach as no such studies have been reported, showing a need for further improvement and facilitation.

The high number of studies concerning flap survival and ischaemia shows that genetic modifications of flaps have been used primarily in this context. Our recent meta-analysis [98] has shown that VEGF delivery, in particular, significantly increases the survival rate of the flap with no significant toxicity in rats. Furthermore, several studies exceed the ischaemia context, focusing on assessing flap survival in other conditions and showing positive outcomes in limiting irradiation damage and allotransplant rejection. Specifically, the second group of studies is promising, as the positive results achieved with combination therapy—OX40 Ig with CTLA4 Ig via viral delivery—may be further expanded to vascular composite allografts, where tissue rejection is a major issue [107].

Finally, there was a small set of studies that aimed to add non-existent therapeutic activities to the flap. Therapeutic goals included treating cancer, osteomyelitis, chronic wounds and systemic protein delivery for protein deficiency disorders. Most of the interventions led to better outcomes compared to controls; however, this poorly represented area requires further exploration.

Most concerns related to gene therapy are associated with systemic or unpredictable distant side effects in non-target locations. The optimization studies included in the analysis showed that the off-target activity of genetic modification following flap-based delivery is non-detectable, and, therefore, the modifications can be considered safe. Furthermore, the included studies did not report any major side effects; however, in most studies, long-term complications could not be observed due to a relatively short assessment period, and further research in this context is therefore required.

An important factor to consider is the long term effectiveness of such therapies, as in most studies, the observation period varied substantially depending on the application. Studies aiming at increasing flap survival reported shorter observation periods—typically between seven and fourteen days—while studies where genetically modified surgical flaps were used as a means to treat other diseases had a much larger observation period margin—up to fourteen weeks. The optimization studies have shown that long term efficacy can depend on the gene delivery method, with intravascular flap perfusion having the strongest and longest-lasting effect. Such an approach was also used in a study by Than et al. [16], which focused on systemic drug delivery using genetically modified flaps, where intravascular perfusion of a free flap had the highest and most long-term efficiency. Hence, it seems that intravascular perfusion should be a method of choice when considering the long-term effectivity of flap transfection/transduction. However, such conclusions were not supported by our recent meta-analysis [98]. Furthermore, another important issue, which is necessary to address before clinical translation, is the establishment of an optimal vector strategy. The included studies used a wide range of vectors, most commonly viruses or plasmids. It seems that when the aim is a prolonged expression, a viral vector may be a better choice, while for short-term expression, high production plasmids may be sufficient. Another strategy is the delivery of transfected or transduced cells; most studies used fibroblasts and mesenchymal-stem cells, which may be interesting vectors due to their high availability from the stromal-vascular fraction (SVF) derived from adipose tissue [108]. Thanks to direct gene therapy of SVF, such a procedure could potentially be undertaken in less-specialized clinics with limited access to laboratories. Although a wide range of vectors have been used, it is difficult to point towards a certain option, as the heterogeneity of studies was very high, and many reported insufficient information regarding the used vector and the durability of the chosen therapy. Overall, it is not possible to lean towards a specific vector strategy as each has its advantages and disadvantages, and the decision should depend on the aim of the therapy.

It is difficult to determine what constitutes biologic brachytherapy; however, we believe that, thanks to comprehensively describing all aspects of flap-based gene therapy, we can develop a common ground for future research. The collection of the studies was not straightforward, as many of them did not have an appropriate title, abstract or keywords leading to a loss of such studies from the search engine. Therefore, we suggest the popularization of a two-word keyword, “biologic brachytherapy”, to increase the detectability of further experiments. The studies included in this review had a wide range of quality—many of them had a high risk of bias—primarily due to not reporting the number of animals, animal losses during the experiment, lack of randomization and blinding. Furthermore, as the applications varied from those related to quickly-assessed flap survival to parameters measured after a longer time period—animal survival or bone regeneration, there was a lack of unified approach and flap models used. Verification of biologic brachytherapy using common flap types, such as random pattern skin flaps, is necessary for comprehensive clinical translation. As in the included studies, either free flaps or pedicled ones were most typically used; therefore, further extension to random-pattern cutaneous flaps and regional or local flap surgeries, which are the most commonly used, is indicated. It is also crucial to consider the replicability of the methodologies described in the research papers, as it was often very challenging to ascertain the specific gene vectors utilized and the combinations that functioned as controls. This ambiguity undermines the validity of such studies, as it is impossible to determine the actual efficacy of the experimental therapies described.

This review has shown that biologic brachytherapy can be applied in a variety of contexts; however, only a dozen studies considered using genetically enhanced flaps to treat secondary conditions and not improve the flap itself. We believe that the next steps should include determining the long-term efficacy of such therapies in more clinically applicable animal models. For modelling breast cancer, as breast-conserving techniques are becoming an important treatment regimen, potentially leading to higher rates of recurrence due to residual disease [109], unless followed by adjuvant radiotherapy, more focus should be placed on residual disease models. Moreover, as only two studies with highly positive results explored the potential of using genetically modified flaps for systemic drug delivery, expanding this branch to include studies on metastatic cancer or diabetes could bring more knowledge on its effectiveness and applicability.

Overall, the included studies show that “biologic brachytherapy”, which aims to combine the reconstructive and therapeutic effect using gene therapy in flap surgery, is effective in many preclinical models. The strongest evidence supports the use of gene therapy to promote flap survival, but other applications are also being developed. Further research is required for the translation of genetically engineered surgical flaps into the clinical setting.

## 5. Conclusions

Biologic brachytherapy uses genetically modified surgical flaps to produce a chosen molecule and exert desired effects on the flap itself, local surroundings or in the whole organism.Biologic brachytherapy, in a preclinical setting, is a feasible therapeutic option for many ailments, including wounds, bone defects, cancer or protein-deficiency disorders. Additionally, they can be used to increase tissue survival in ischemic flaps, but also in other contexts, such as irradiation-damage management, and in allotransplants to reduce the immune response.Biologic brachytherapy of flaps with VEGF appears to be well-grounded for first-in-human studies. Other therapeutic aims require more research.New investigations should focus on high-quality data reporting, studying long-term side effects, expanding the use of genetically enhanced flaps for systemic therapies and determining the optimal transgene delivery method.

## Figures and Tables

**Figure 1 ijms-25-10330-f001:**
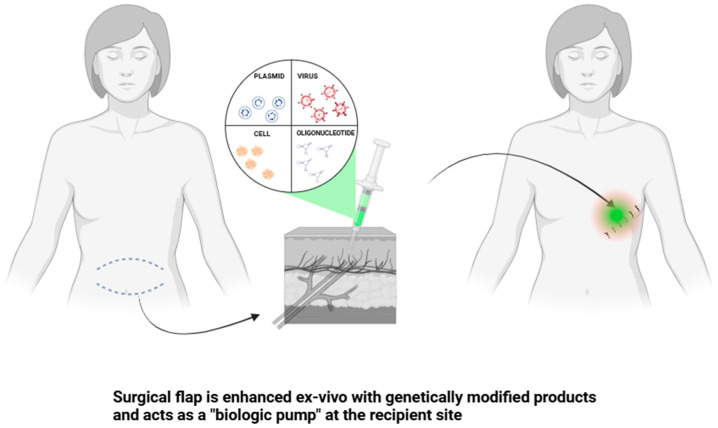
Figure presenting the basic idea of biologic brachytherapy. Created with BioRender.com.

**Figure 2 ijms-25-10330-f002:**
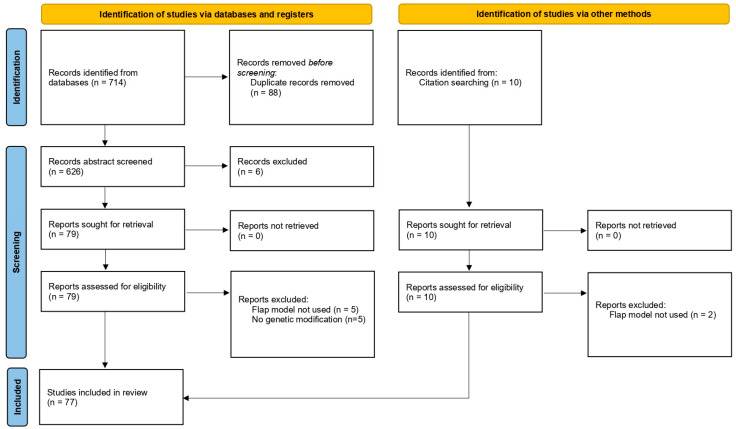
PRISMA flowchart of included studies.

**Figure 3 ijms-25-10330-f003:**
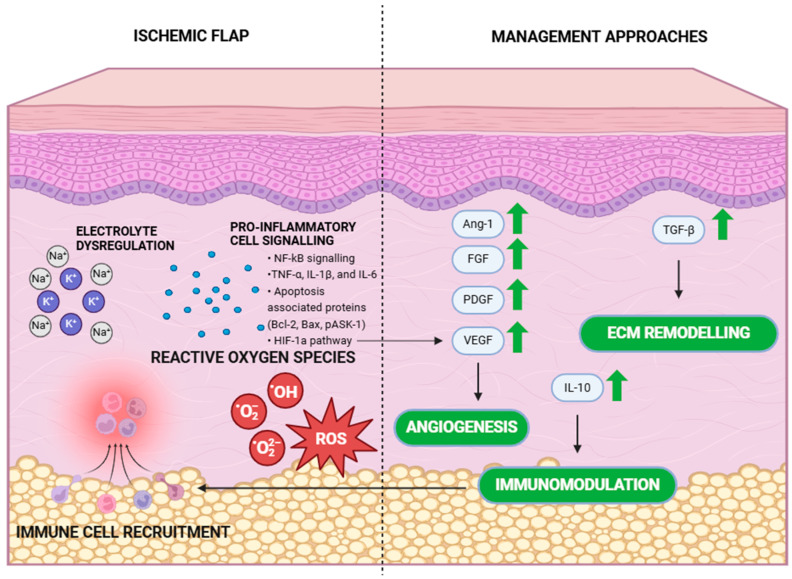
Figure showing pathological processes in ischemic surgical flap and currently developed management approaches using genetically enhanced surgical flaps. Green arrows show the main treatment strategies using target upregulation and black arrows show the effects of such treatment. Created with BioRender.com.

**Figure 4 ijms-25-10330-f004:**
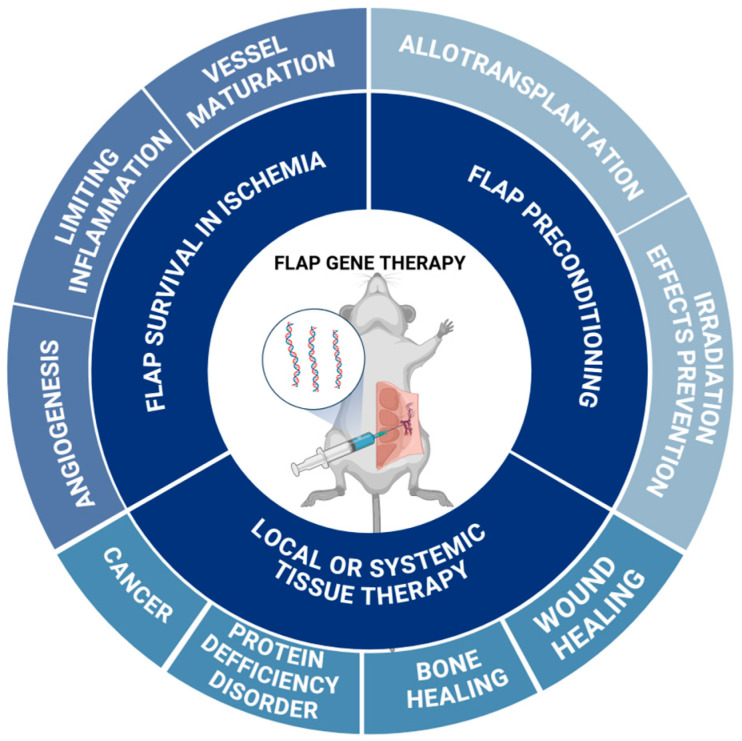
Figure presenting the summary of potential conditions treated with biologic brachytherapy tools. Created with BioRender.com.

**Table 1 ijms-25-10330-t001:** Overview of studies that used surgical flaps for local or systemic tissue therapy.

1st Author; Year	Aim of the Therapy	Target Gene/s	Vector	Surgical Technique	Experimental Groups	Method(s) of Verification	Observation Period	Flap Model	Number of Animals and Species	Final Results of Therapy (* = Not Significant)
Casal et al., 2019 [9]	Increasing healing of Pseudomonas-infected wounds	BD-2 BD-3	Virus	Viral titre inoculated intraarterially. *P. Aeruginosa* injected to induce infection and clamped for ninety minutes. All groups received VEGF at the time of injection. Foreign body was implanted to produce foreign body response.	(A) NaCl-treated group (B) *P. Aeruginosa*-infected only group (C) GFP-treated group (D) Len-BD-2-treated group (E) Len-BD-3-treated group (F) Len-BD-2 and Len-BD-3-treated group	Flap survival, gene expression and bacterial cell count	7 days	Fasciocutaneous flap	102 rats	Increase in flap survival in groups (A/D/E/F) compared to groups (B/C). Number of bacteria in catheter lowered in group (A) when compared to group (C).
Ghali et al., 2009 [10]	Increasing healing of chronically infected wounds	LL37	Virus	Prior to flap infusion with viral titre, bacteria inoculated to produce chronic wound infection.	(A) Ad/CMV-LL37-treated group (B) Ad/CMV-LL37 and VEGF-treated group (C) Ad/CMV-LacZ-treated group (D) PBS-treated group	Bactericidal activity	7 days	Superficial inferior epigastric fasciocutaneous free flap	26 rats	Increase in bactericidal activity in groups (A/B) when compared to groups (C/D).
Saad et al., 2018 [11]	Increasing healing of ischemic wounds in a flap	Trx-1	Virus	Performed with two skin pedicles. Punch biopsy used to create a full thickness wound in the middle of ischemic skin flap, followed by two non-ischemic wounds on either side of flap; metal splint was inserted in periwound skin. Four intradermal injections were performed in each wound in four quadrants. Groups (A/B/C) on ischemic wounds and groups (D/E) on non-ischemic wounds.	(A/D): Ad-Trx-1-treated group (B/E): Ad-LacZ-treated group (C/F): Control without gene therapy group	In vitro scratch assay on HUEVOC cells. In vivo wound closure rate. TUNEL, immunofluorescence, Immunohistochemistry Trx-1 and capillary density	9 days	Modified McFarlane flap	No clear number of mice	Percentage of wound closure in groups (D/E/F) not significantly different than non-ischemic wounds. Mean wound closure is higher in group (A) when compared to group (B). Vessel density highest* in group (A). Ki-67 expression increased in group (A) compared to groups (B/C), also in group (B) compared to group (C). At 24 h, wound closure increased in group (A) compared to groups (B/C).
Lampert et al., 2015 [12]	Increasing healing in critical-size bone defects	BMP-2	Virus	Muscle was harvested and critical-size defect was created. Next, the muscle was perfused with titre dependent on the group.	(A) Ad-BMP-treated group (B) Ad-GFP-treated group (C) Saline-treated group	Increase in bone regeneration and protein synthesis	14 weeks	Quadriceps femoris muscle flap	12 rats	Effective transduction and local increase of BMP-2. Increase in bone density at 14 weeks in groups (A) and (B) when compared to group (C).
Dempsey et al., 2008 [14]	Treatment of breast cancer	IL-12	Virus	Intraarterial. Vector was incubated within a flap for one hour ex vivo and transplanted back to its original position in the groin over a bolus of MADB106 cells.	(A) Ad/RSV-mIL-12-treated group (B) Ad/CMV-lacZ-treated group (C) PBS-treated group (D) Ad-IL-12 by tail vein-treated group	IL-12 production and tumour growth suppression	28 days	Fasciocutaneous superficial inferior epigastric flaps	32 rats	Tumour volume on day 27 decreased in group (A) compared to group (B). Serum bHCG decreased in group (A) when compared to group (B). Local expression of IL-12 corresponded with a 4.5-fold increase in IFN-γ expression when compared with group (A).
Seth et al., 2015 [15]	Treatment of residual disease (MiRD and MaRD)	TK	Virus	In experiment one, flap elevation and induction of MiRD with RG2 cells. In experiment two, induction of MaRD by leaving RG2 cells covered with flaps and perfused with treatment. (A/D) had Ad.TK MOI of 50 and (B) had Ad.TK MOI of 10.	Exp.1: (A/B): Ad-TK and ganciclovir-treated group (C) PBS and ganciclovir-treated group Exp.2: (D) Ad-TK and ganciclovir-treated group (E) PBS and ganciclovir-treated group	Treatment efficacy in MiRD and MaRD, systemic viral distribution and gene expression	~48 days	Epigastric flap	36 rats	Median time to achieve palpable tumour lowered in groups (A/B) when compared to group (C). Median survival time is longer in groups (A/B) when compared to group (C). Median delay to recurrence was increased in group (D) when compared to group (E). TK mRNA expression higher in groups (A/B) when compared to group (C). mRNA expression in cephalic flap tissues and the SIEA pedicle was greater in flaps infected at an MOI 10 than those infected at MOI 50.
Zhang et al., 2019 [17]	Treatment of breast cancer	CDglyTK	Virus	SHZ-88 cells subdermally injected before flap ex vivo perfusion. During the ex vivo period, afferent artery to the flap was catharized with a micro-cannula and treatment solution. PBS flushing was performed pre- and post-op. 5-Flucytosine and ganciclovir administered to all groups until 21 days post-op.	(A) CDgly-TK-treated group (B) CDgly-GFP-treated group (C) PBS-treated group	Tumour volume, numerous in vitro assays, CDglyTK in flap and systemic expression	42 days	Epigastric flap	18 rats	Mean tumour volume and tumour weight decreased in group (A) when compared to groups (B/C). At 15 days and 42 days, after SIEA flap transfection, the protein expression of the CD/TK fusion gene was undetectable using immunohistochemical staining and Western blot assay.
Davis et al., 2020 [18]	Treatment of breast cancer	IFNγ	Virus	After flap elevation, flap artery was cannulated and flap was irrigated with PBS. Secondly the vein was clamped, and viral vector was injected via arterial cannula with one hour dwell time. After washing and vascular repair, MADB-106-Luc or MAD-MB-231 cells were injected deep to the flap.	(A) AAV-DJ-CMV- IFNγ-treated group (B) AAV-DJ-CMV-GFP-treated group	Cancer emission signal, flap survival, histological analysis, subject survival, IVIS signal, IFNγ distribution local or systemic	28 days	Microvascular free flap	20 rats	From 5 days onward, decreased levels of breast cancer cells in group (A) compared to group (B). Cancer cell cell luminescence decreased in group (A) compared to group (B). Histopathological slide of group (A) revealed tissue that develops from tumour undergoing regression or destruction. Survival increased in group (A) compared to group (B). IFN-γ concentration in group (A) was increased locally when compared to systemic.
Michaelis et al., 2004 [7]	Systemic production of anti-angiogenic protein	Endostatin	Virus	Viral mixture perfused through the femoral artery, with femoral vein being clamped. After a one hour dwell period in a 37 °C humidified incubator, the flap was flushed with PBS.	(A) Ad5/CMV-mENDO-treated group (B) Ad5/CMV-lacZ-treated group	Serum endostatin and local endostatin expression	10 days	Quadriceps muscle free flap	12 rats	Elevation* in serum endostatin from day 3 until the end observation. Average serum levels of endostatin were elevated in group (A) compared to (B). No endostatin was observed in group (B).
Aliyev et al., 2016 [13]	Increasing bone healing using VEGF therapy in osteomyelitis	VEGF-165	Plasmid	Osteomyelitis induced in tibial bone. Afterwards, depending on group, treatment was given for four weeks. After muscle elevation, plasmid was injected into the muscle at three time points.	(A) No treatment group (B) Gentamicin-treated group (C) Normal muscle flap and gentamicin-treated group (D) VEGF plasmid transfected muscle flap and gentamicin-treated group	WBC count, body temperature, histological and radiological control of abscess	28 days	Medical gastrocnemius muscle flap	32 rats	Body temperature was lower in group (D) compared to groups (A) and (B). Lowest* WBC count in group (C) and highest* in group (A). Total radiographic score and histological score were lowest* in group (D) and highest* in group (A).
Than et al., 2022 [16]	Treatment of haemophilia B	hFIX	Minicircle	In Exp 1, murine inguinal fat pads were injected with minicircle. In (A), minicircle concentration was 10 µg/µL, while in (B), it was 30 µg/µL. In Exp 2, SIAE flaps were elevated and transfected ex vivo using intraarterial injection with minicircle. After a one hour dwell time, vein unclamped, flashed with PBS and reimplanted back.	Exp 1 (A/B) Minicircle DNA luciferase-treated group (C) PBS-treated group Exp 2 (D) Minicircle DNA luciferase-treated group (E) Minicircle DNA hFIX-treated group (F) PBS-treated group	Bioluminescence, hFIX ELISA	60 days in Exp 1 and 28 days in Exp 2	SIEA flaps	9 rats in Exp 1 and 10 rats in Exp 2	At day 60 in Exp 1, Group (A) showed persistent bioluminescence; group (B) decreased to ground level. In Exp 2, hFIX within systemic circulation and locally was higher in group (E) in comparison to group (F).

BD—beta defensins, Len—lentivirus, GFP—green fluorescent protein, LL-37—human cathelicide in antimicrobial peptide, Ad—adenovirus, LacZ—beta-galactosidase, Trx—Thioredoxin, CMV—cytomegalovirus, BMP—bone morphogenic protein, VEGF—vascular endothelial growth factor, IL-12—interleukin 12, MADB106—breast cancer cells, RSV—respiratory syncytial virus, bHCG—beta human chorionic gonadotropin, IFNγ—interferon gamma, MaRD—macroscopic residual disease, MiRD—microscopic residual disease, TK—thymidine kinase, RG2 cells—rat glioma cells, MOI—multiplicity of infection, PBS—phosphate buffer saline, SIAE—superficial inferior epigastric artery, CDglyTK—cystosine deaminase-thymidine kinase, SHZ-88—breast cancer cells, post-op—post-operatively, AAV—adeno-associated virus, MADB-106-Luc—breast cancer cells with luciferase, MAD-MB-231—breast cancer cells, IVIS—in vivo imaging systems, hFIX—human factor IX.

**Table 2 ijms-25-10330-t002:** Overview of studies focusing on flap preconditioning affecting surrounding tissue and the flap.

1st Author; Year	Aim of the Therapy	Target	Vector	Surgical Technique	Experimental Groups	Outcome(s)	Observation Period	Flap Model	Number of Animals and Species	Final Results of Therapy (* = Insignificant)
Khan et al., 2018 [28]	Mitigating radiotherapy effects	SOD2 CTGF	Virus	Intraarterial injection was performed and left to incubate for 50 min before the vascular compartment was flushed through and vascular anastomoses performed. Breast carcinoma cells (1 × 10^7^) were injected at 26 days post-op directly into flap and monitored daily for tumour growth. Flap irradiation was performed for 5 days (days 27–31) post-op.	(A) Not irradiated group (B) PBS-treated group (C) LV-SOD2+-treated group (D) LV-shCTGF+-treated group (E) LV-Scramble-treated group	RTOG score, gene expression, flap surface area, MRI flap control, immunohistochemistry and histological analysis	180 days	SIEA flap	30 rats	(C) and (D) showed greatest results in preservation of flap volume or reduction in skin contracture.
Fu et al., 2010 [30]	Preventing flap rejection by delivering immunomodulatory molecule	OX40Ig	Virus	Intraarterial injection of experimental solutions was administered. Then, orthotopic SIEA flap transplantation from Brown Norway to Lewis rats was performed.	(A) Control group (B) Ad.EGFP-treated group (C) Ad-OX40Ig-treated group (D) Ad-OX40Ig and rapamycin-treated group (E) Rapamycin-treated group	Gene expression and flap survival time	7 days	SIEA flap	45 rats	Flap survival was greatest in (D) when compared with other groups.
Xiao et al., 2011 [31]	Increasing allotransplanted flap survival	CTLA4Ig	Virus	Intraarterial injection of respective solutions was administered. Flaps were perfused ex vivo with pump, 5 mL PBS, then 1 mL adenoviral solution and then 2 mL PBS. In group (A/C), treatment was performed 6 days pre-op. Rapamycin in those groups was given at 2 mg/kg concentration.	(A) Ad-CTLA4Ig and rapamycin-treated group (B) Ad-CTLA4Ig-treated group (C) Ad-GFP-treated group (D) Rapamycin-treated group (E) Non-treated group	Banff classification, gene expression and mixed lymphocyte reaction	62 days	Epigastric flap	32 rats	Flap survival was the longest in (A) when compared with other groups. Banff grade III/IV rejection in (C/E) at day 7, Banff grade III at day 15 in (B). Normal skin histology at day 15 of (A) and almost intact at day 30. Lymphocyte reaction was the lowest in (A) when compared with other groups. Cytometry revealed significant increase in CD4 + 25 + Foxp3+ T cells in (A/D) vs. (E), but not in (B/C) vs. (E)
Zhang et al., 2012 [32]	Increasing allotransplanted flap survival	OX40I CTLA4Ig	Virus	Intraarterial injection of respective solutions was administered. Flaps were perfused ex vivo with pump, 5 mL PBS, then 1 mL lentiviral solution and then 2 mL PBS. In groups all groups except control, treatment was performed 7 days pre-op.	(A) Len-OX40Ig, CTLA4Ig and rapamycin-treated group (B) Len-CTLA4Ig and rapamycin-treated group (C) Len-OX40IIg and rapamycin-treated group (D) Rapamycin-treated group (E) Len--EGFP-treated group (F) Non-treated group	Animal survival, gene expression and mixed lymphocyte reaction	63 days	Epigastric flap	35 rats	Animal survival was the highest* in (A) when compared with other groups. On day 28, (D/E/F) flaps were necrotic/dead, and (A/B/C) showed mild inflammatory infiltration. Graft (A) was “almost intact”. Mixed lymphocyte reaction was lowest in (A) when compared with other groups. Serum cytokine IL-2 was the lowest in (A) when compared with other groups. Serum cytokines IL-4 and IL-10 were the highest in (A) when compared with other groups.
Angelos et al., 2011 [29]	Improving revascularization and irradiated flap viability	VEGF-165	Plasmid	Irradiation was performed. Then a 28-day recovery period after which a vascular clip was applied for 2 h to stimulate an ischaemic period. Topical VEGF pDNA, in vivo cationic polymer (jetPEI) and fibrin sealant were then administered during flap elevation. Pedicles in each group were ligated in the respective time periods—day 8 (A/C) and day 14 (B/D).	(A) VEGF pDNA as topical cationic polymer and fibrin sealant-treated group (B) VEGF pDNA as topical cationic polymer and fibrin sealant-treated group (C) Topical cationic polymer and fibrin sealant-treated group (D) Topical cationic polymer and fibrin sealant-treated group	Flap revascularization and flap viability	5 days	Ventral fasciocutaneous flap	28 rats	Flap revascularization was greatest in (B) when compared with other groups.
Jeong et al., 2024 [33]	Increasing allotransplanted flap survival and miRNA expression	IL-10	Plasmid	Intraarterial injection with vein cross-clamped at the pedicle was performed. Sprague Dawley rat flap was perfused ex vivo and subsequently transplanted to Wistar rats.	(A) Ad-IL-10 plasmid-treated group (B) Saline-treated group	Flap survival, miRNA expression, histological analysis and immunohistochemical staining	7 days	Epigastric flap	No clear number of rats	Flap survival was improved, and acute rejection response was reduced* in (A) when compared with control (B). On seventh day post-op, IHC and RT-PCR revealed positive IL-10 expression in (A). Expression of target miRNAs for skin tissue and target miRNAs for serum were both positive for (A).

RTOG—radiation therapy oncology group score for radiation toxicity grading, MRI—magnetic resonance imaging, SIEA – superficial inferior epigastric artery, SOD2—superoxide dismutase 2, CTGF—connective tissue growth factor, LV – lentivirus, sh – short hairpin RNA, OX40Ig - OX40 immunoglobulin, CTLA4Ig - cytotoxic T-lymphocyte-associated protein 4 immunoglobulin, Ad – adenovirus, VEGF—vascular endothelial growth factor, CTLA—cytotoxic T-lymphocyte associated protein, EGFP—enhanced green fluorescent protein, IL-10 – interleukin 10, RT-PCR—reverse-transcription polymerase chain reaction, IHC—immunohistochemistry.

**Table 3 ijms-25-10330-t003:** Overview of studies that focused on increasing ischemic flap survival.

**1st Author; Year**	Aim of the Therapy	Target Gene/s	Vector	Surgical Technique	Experimental Groups	Method(s) of Verification	Observation Period	Flap Model	Number of Animals and Species	Final Results of Therapy (Non-Significant = *)
Meirer et al., 2007 [43]	Increasing flap survival and blood supply to the flap	VEGF	Virus	The right inferior epigastric artery and vein were left intact, whereas the left inferior epigastric vessels were ligated and divided. The proximal border of the flap was incised to create a skin island flap pedicled on the right inferior epigastric vessels. Subdermal injections were administered into seven spots.	(A) Shockwave-treated group (B) Ad-VEGF-treated group (C) Control group	Area of necrotic zone	7 days	Epigastric skin model	30 rats	Flap survival was the highest* for group (A). Area of necrotic zone was decreased for groups (A/B) when compared to group (C), and group (A) when compared to group (B).
Lubiatowski et al., 2002 [45]	Increasing flap survival and flap revascularization	VEGF	Virus	Two days prior to flap elevation, subdermal injections were administered. Flap viability was evaluated at day seven and fourteen after procedure. In (A/B), the virus was administered locally to the predicted area of necrosis. In (C/D), the virus administered distally from area of ischaemia, midline axis of flap group.	(A/C) Ad-VEGF-treated group (B/D) Ad-GFP-treated group (E) No transfection prior to flap elevation group	Percentage of necrotic and hypoxic tissue and neovascularization	14 days	Epigastric skin flaps based on inferior epigastric vessels	30 rats	At day 14, percentage of necrotic tissue was decreased for groups (A/C) when compared to groups (B/D/E). At day 14, percentage of combined necrotic/hypoxic zones was decreased* for groups (A/C) when compared to groups (B/D/E). There were no significant differences in neovascularization among groups.
Gurunluoglu et al., 2002 [50]	Increasing angiogenesis of flap	ANG-1	Virus	Substance administered as intra-arterial injection.	(A) PBS-treated group (B) Ad-ANG-1-treated group (C) Ad-GFP-treated group	Capillary density	14 days	Cremaster muscle tube flap	45 rats	The number of flowing capillaries was increased in group (B) when compared to (A/C). Microvascular permeability index was increased in group (C) when compared to group (B).
Huang et al., 2006 [52]	Increasing skin flap viability, synthesis/release of angiogenic and vasodilatory factors	VEGF-165 eNOS	Virus	Substances subdermally injected into the distal half of the skin flap seven days before surgery. The injections were spaced half a centimetre apart along both sides of the midline and at one centimetre from the midline.	(A) PBS-treated group (B) PBS and empty Ad-treated group (C) PBS and Ad-VEGF-165-treated group (D) PBS and Ad-eNOS-treated group	Flap viability, capillary density, protein expression and effect of inhibition of indomethacin	7 days	Dorsal random-pattern skin flap	24 rats	Flap viability was shown to be dose dependent in group (C). There was also an increase in flap survivability in group (C) when compared to group (D). Capillary density was increased in group (C) when compared to group (A/B). Skin blood flow 9 h post op was increased in group (C) when compared to group (A/B).
Lubiatowski et al., 2002 [54]	Increasing flap perfusion	VEGF-165 ANG-1	Virus	Intraarterial injections were performed at the time of flap elevation. External iliac vein was clamped distally and proximally to the pedicle, micro clamps were removed after 15 min.	(A) PBS-treated group (B) Ad-GFP-treated group (C) Ad-VEGF-treated group (D) Ad-ANG1-treated group (E) Ad-VEGF and Ad-ANG1-treated group	Functional capillary density and microvascular permeability index	14 days	Cremaster muscle flap model	90 rats	On day 7, functional capillary density was increased in groups (C/D/E) when compared to group (A/B). On day 14, perfused capillary count was increased in groups (C/D/E) when compared to group (A/B).
Rah et al., 2014 [59]	Increasing flap survival	HGF	Virus	Eight injections were made into the subdermal layer of the entire area of the skin flap two days before flap elevation and immediately after flap elevation.	(A) Ad-HGF-treated group (B) Recombinant HGF-treated group (C) PBS-treated group	Survival area of flap, ratio of blood flow, CD31-positive vessel counts and VEGF expression	10 days	Dorsal skin flap with panniculus carnosus	30 rats	Flap survival increased in group (A) when compared to groups (B/C). Ratio of blood flow in mid-distal flap was increased in group (A) at days 7 and 10 compared to groups (B/C). In distal flap, there was an increase in the blood flow ratio on day 3 and 7 in group (A) compared to groups (B/C).
Huemer et al., 2004 [60]	Increasing the survival of ischaemic flaps by increasing the angiogenic potential	TGF-β	Virus	Just prior to flap elevation, the injections were given subdermally in the left upper corner of the flap.	(A) Ad-TGF-b-treated group (B) Ad-GFP-treated group (C) 0.9% NaCl-treated group	Mean percent surviving area and neovascularization by immunohistochemistry	7 days	Epigastric skin flap based on inferior epigastric vessels	30 rats	Flap survival increased in group (A) when compared to group (B/C). Increase* in capillary number in group (A) compared to groups (B/C).
Huemer et al., 2005 [61]	Increasing neoangiogenesis in failing flaps	TGF-β	Virus	Injections were made into the subdermal space, with seven points into the left upper corner of the flap, just before flap elevation.	(A) Ad-TGF-b-treated group (B) ESW-treated group (C) No-treatment group	Mean percent surviving area, angiogenesis by CD31 immunohistochemistry and histologic evaluation	7 days	Epigastric skin flap based on inferior epigastric vessels	30 rats	Flap survival increased in groups (A/B) compared to group (C) and in group (B) when compared to group (A). Increased* number of capillaries in (A/B).
Liu et al., 2009 [62]	Increasing survival of ischaemic flaps, viability and vascularization	bFGF	Virus	Viral particles were intradermally injected to the dorsum of rats. In groups (A/D), flap elevation took place immediately after viral injection; in group (B), it took place one week after; and in group (C), two weeks after.	(A/B/C) pAAV2/CMV-bFGF-treated group (D) Saline-treated group	Surviving area of flap, neovascularization and bFGF gene expression	7 days	Random skin flaps	38 rats	Flap survival was increased in group (A) when compared to group (D) and in group (D) when compared to group (B). Based on histological evaluation, best* vasculature was viewed in group (C).
Lee et al., 2011 [64]	Increasing flap survival	RLX	Virus	Two days before and immediately after flap elevation, viral mix was injected subdermally.	(A) dE1-RGD/lacZ-RLX-treated group (B) dE1-RGD/lacZ-treated group (C) PBS-treated group	Flap survival, flap blood flow and capillary density	10 days	Dorsal skin flap including panniculus carnosus	30 rats	Flap survival was increased in group (A) when compared to other groups (B/C). Vascular flow was increased at day 10 post-op for group (A) when compared to all the other groups (B/C). The number of CD31-positive blood vessels was the highest* in group (A).
Jung et al., 2003 [65]	Increasing postoperative flap survival	ANG-1	Virus	Two days before flap elevation, seven points of subdermal injections with viral mixture were made into the left upper corner of the flap.	(A) Ad-ANG-1-treated group (B) Ad-GFP-treated group (C) No injection group	Percentage of necrotic area	7 days	Epigastric skin flaps	19 rats	Flap survival was increased in group (A) when compared to other groups (B/C). Vascularity was increased* in group (A) compared to all the other groups (B/C).
Choi et al., 2020 [66]	Increasing survival and angiogenesis of the flap	DKK2	Virus	Two days before and immediately before flap elevation, subdermal injection, evenly made in twelve injection points per flap.	(A) dE1-RGD/DKK-treated group (B) dE1-RGD-treated group (C) PBS-treated group	Flap survival rate, cutaneous blood flow, CD31 and VEGF staining	10 days	Random-pattern dorsal cutaneous flap	30 rats	Flap viability was increased in group (A) when compared to group (C). In compartment 3, after days 7 and 10 post-op, perfusion was increased in group (A). The number of CD31-positive blood vessels was increased in group (A) when compared to group (C).
Lou et al., 2021 [67]	Increasing overall flap survival by inhibiting PLA2G4E to restore lysosomal function and inhibit flap necroptosis	PLA2 G4E	Virus	Part I of the experiment was performed after flap elevation. During part I, authors extracted tissue proteins from each flap group for further analysis. Then, 28 days prior to flap elevation, subcutaneous injections of viral particles in three predetermined flap areas were administered in both part II and III.	(A) No treatment group Part I: (B) Oral saline-treated group (C) Oral NDI-treated group Part II: (D) Saline-treated group (E) NDI-treated group (F) AAV-Pla2g4e shRNA and PBS-treated group (G) AAV-Pla2g4e shRNA and NDI-treated group Part III (H) Saline-treated group (I) AAV-scramble-I-treated group (J) AAVPla2g4e shRNA-treated group (K) AAV-scramble-II group-treated group (L) AAV-Mir504-5pup-treated group	Lysosomal markers analysis, PLA2G4E expression, lysosomal membrane permeabilization change, immunohistochemistry, collagen damage and cell death	7 days	Random-pattern skin flap	191 mice	Survival area increased in group (J) when compared to groups (H/I). Signal intensity of blood flow was increased in group (J) when compared to groups (H/I).
Gurunluogu et al., 2002 [68]	Increasing flap viability and vascularity	VEGF-164	Virus	Three hours to fourteen days prior to flap elevation, subdermal injections with viral mixture were administered. Each group was subdivided based on different times of injection before flap elevation (12 h, 3 days, 7 days and 14 days).	(A) Control group (B) Ad-GFP-treated group (C) Ad-VEGF-treated group	Percentage of skin necrosis and vascularity	7 days	Epigastric island flap	84 rats	Necrotic flap area was decreased in group (C) across all measured time points when compared to other groups (A/B). Density of vessels appeared increased* in all subgroups of (C) when compared to (A) and seemed to be identical in all (C) subgroups.
Gurunluogu et al., 2005 [69]	Increasing flap viability and vascularity	VEGF-121	Virus	Subdermal injections with viral mixture were performed.	A) Saline-treated group (B) Ad-GFP-treated group (C) Ad-VEGF-treated group	Percentage of skin necrosis and vascularity	7 days	Peninsular abdominal flap based caudally	34 rats	Flap survival increased in group (C) when compared to group (A/B).
Giunta et al., 2005 [70]	Finding the perfect viral concentration to increase flap survival	VEGF-165	Virus	Subcutaneous injections with viral mixture at different times before and during flap elevation. Groups (A/B/E/F/G/H) were injected seven days prior; groups (D), three days prior; and group (C), zero days prior. Same viral concentration of 5 × 10^8^ pfU in groups (C/D/E).	(A) 0.9% NaCl-treated group (B) Ad3/12-treated group (C/D/E): Ad-VEGF-treated group (F) Ad-VEGF (1 × 10^9^ pfU)-treated group (G) 0.9% NaCl with no flap elevation group (H) Ad-VEGF (1 × 10^9^ pfU) with no surgery group	Percentage of skin necrosis, perfusion maximum and perfusion index	7 days	Abdominal, random pattern flap model	50 rats	Flap survival and perfusion index increased in groups (D/E/F) when compared to group (A).
Antonini et al., 2007 [76]	Reducing flap necrosis	VEGF-165	Virus	Different flaps were used—epigastric (A–F), musculocutaneous (G–L). Treatment for each group was given at the time of flap elevation (A/D/G/J), seven days prior (B/E/H/K) or fourteen (C/F/I/L) days prior.	(A/B/C) Epigastric, VEGF groups (D/E/F) Epigastric, LacZ groups (G/H/I) Musculocutaneous, VEGF groups (J/K/L) Musculocutaneous, lacZ groups	Flap necrosis and neovascularization	14 days	Epigastric cutaneous flap, composite musculocutaneous flap	48 rats	Most effective was a musculocutaneous injection 7 and 14 days prior (I/H) compared to other groups. There was also a significant reduction in flap necrosis in those groups (I/H) compared to other groups.
Taub et al., 1998 [78]	Increasing survival of ischaemic experimental skin flaps	VEGF	Virus	During right-sided flap elevation and subsequent saline wash of the flap, therapeutic and control solutions were injected into the femoral artery distal to the origin of the epigastric artery, with a dwell time of 10 min. Four days after flap creation, the pedicle was ligated. Background fluorescence was measured at four different sites at three different time points.	(A) pAd-MCS-VEGF-treated group (B) pAd-MCS-treated group (C) Saline-treated group	Dye fluorescence index, percentage of viable tissue, VEGF expression and histological staining	7 days	Axial pattern skin flap	30 rats	Dye fluorescence index 2 h after treatment was the highest in group (A). Flap survival was increased for group (A) in comparison to group (C). The average mean number of blood vessels was increased* in group (A) compared to group (C). Mean lumen diameter of vessels in group (A) was lower than in group (C).
Zacchigna et al., 2005 [82]	Increasing flap survival	VEGF-165	Virus	Ten equally spaced subcutaneous VEGF solution (A/B/C) and LacZ solution (D/E/F) direct injections in epigastric flap, which were done at the time of flap elevation (A/D), 7 days prior (B/E) and 14 days prior (C/F). Based on the result of the first experiment, additional rats were intramuscularly injected with VEGF solution (G/H/I) and LacZ solution (J/K/L) into the TRAM flap at the time of flap elevation (G/J), 7 days prior (H/K) and 14 days prior (I/L/AA/BB).	Exp. 1 (A/B/C/G/H/I) Ad-VEGF-treated groups (D/E/F/J/K/L) Ad-LacZ-treated groups Exp. 2 (AA) Ad-VEGF-treated group (BB) Ad-lacZ-treated group	Flap necrosis and HE assessment (second exp.)	Exp. 1: 7 days Exp. 2: 7 days	Epigastric flap TRAM flap	88 rats	In Experiment 1, overall greatest results for increase in flap survival were noted for group (I) when compared with other groups. In Experiment 2, best results were observed in group (AA). In histological semiquantitative analysis, (AA) flaps showed improved skin tissue quality (total score 11.9 vs. 6.3 in (BB) vs. 14.8 in normal skin). After assessment of number of CD31 vessels, (AA) group showed greater* numbers when compared with (BB).
Wang et al., 2011 [90]	Increasing flap survival	VEGF-165	Virus	Fourteen days before flap elevation, 21 intradermal injection, each 100 μL.	(A) AAV-VEGF-treated group (B) AAV-GFP-treated group (C) Saline-treated group	Flap survival, histology and gene expression	7 days	McFarlane flap	30 rats	Flap survival was the greatest in group (A) when compared with other groups. Improved vasculature in (A) vs. (B/C). No difference in inflammation. Markedly increased expression of EGF, PDGF, VEGF (3, 3 and 13 folds in (A) vs. (C)), substantial down regulation in FGF 2 expression CXCl2 and MMP9 up regulated in (A) and (B) vs. (C)
Wang et al., 2013 [92]	Increasing healing of ischaemic flaps	KGF	Virus	Multiple subdermal injection (1 mL) in wound margin of necrotic flap. Injected 5 days after flap elevation.	(A) Dexamethasone and Ad-KGF-treated group (B) Dexamethasone and Ad-control-treated group (C) Dexamethasone-treated group (D) PBS-treated group	Flap necrosis, gene expression, histology and IHC (FGF, CD34)	35 days	McFarlane flap	60 rats	Necrotic area was significantly lower on the 15th and 35th day in group (A) vs. (B/C/D). (A) had the thickest epithelium on days 15 and 25 vs. (B/C/D). At 35 days, A,B,C,D had similar epithelium thickness.
Uemura et al., 2012 [91]	Increasing ischaemic flap survival	NF-kB	Synthetic double-stranded oligodeoxynucleotide	Intraarterial (200 μL) injection through contralateral artery.	(A) NF-kB decoy ODN-treated group (B) Single-strand ODN-treated group (C) No injection group	Flap survival and histology—biopsies at 24th hour, TNF-a, IL-1b and IL-6 qPCR expression in biopsies (24th hour) and IHC (iNOS expression)	5 days		36 rats	Flap survival was the greatest in group (A) when compared with other groups. PMNs count was the lowest* in group (A) when compared with other groups. A decreased* expression of TNF-a, IL-1b and IL-6 at the mRNA level in (A) vs. (B/C). IHC and iNOS staining revealed lowest levels for group (A) when compared with other groups (B/C).
Salafutdinov et al., 2021 [40]	Increasing angiogenesis	VEGF-165 FGF-2	Plasmid	Injection of plasmid evenly dispersed in skin flap during flap elevation. During the procedure, the skin flap was clipped.	(A) VEGF-165 and FGF-2 plasmid-treated group (B) NaCl-treated group	Gene expression and blood flow in skin flap	4 days	Dorsal skin-fascial flap	20 rats	Flap survival not measured. Blood flow, after 4 days, increased* in group (A) when compared to group (B).
Neumaister et al., 2001 [41]	Increasing viability of muscle flaps	VEGF-165	Plasmid	Intramuscular injection at the end of a four hour induced ischaemia, produced via clamping of main femoral vessels.	(A) VEGF165 plasmid-treated group (B) Control group	Flap viability and capillary-to-muscle fibre ratio	7 days	Gracilis muscle microcirculation model	12 rats	Both flap survival and capillary per muscle fibre ratio increased in group (A) when compared to group (B).
Michlits et al., 2007 [42]	Protecting flaps against necrosis and increasing angiogenesis	VEGF-A	Plasmid	Prior to flap infusion with viral titre, bacteria inoculated to produce chronic wound infection. Plasmid was fibrin-mediated.	(A) Control group (B) Fibrin sealant locally onto the fascial layer of the recipient bed to which the flap was sutured group (C) Empty plasmid and fibrin sealant-treated group (D) Local fibrin-mediated VEGF plasmid-treated group	Flap survival, flap perfusion and VEGF expression	7 days	Modified epigastric flap model	48 rats	No differences were observed in flap survival and percentage of ischemic zones among the groups. The percentage of necrotic areas on day 3 and 7 decreased in group (D). Flap perfusion on day 3 and 7 increased in group (D).
McKnight et al., 2008 [44]	Increasing flap survival and flap revascularization	VEGF-165	Plasmid	Ischaemia was induced for two hours using vascular Heifitz clip. After the ischaemia, treatment was applied. All rats had buffered solution/VEGF-protein/VEGF-plasmid suspended in the fibrin sealant, topically.	(A) Buffer in fibrin-treated group (B) VEGF protein in fibrin sealant-treated group (C) VEGF-165 plasmid in fibrin sealant-treated group	Percentage of flap survival and neovascularization	7 days	Fasciocutaneous flaps based on inferior epigastric vessels	20 rats	Flap survival was increased in groups (A/B) when compared to group (C).
Liu et al., 2005 [46]	Increasing survival and vascularity of the ischaemic flap	PDGF-B VEGF	Plasmid	Seven days prior to flap elevation, intradermal injections were administered.	(A) PDGF-B plasmid-treated group (B) Saline-treated group (C) Empty plasmid-treated group (D) VEGF plasmid-treated group	Flap viability and neovascularization	7 days	Caudally based random pattern McFarlane flap	45 rats	Flap survival was increased in groups (A/D) when compared to groups (B/C). The density of blood vessels was increased in group (A) when compared to group (C).
Freitas et al., 2010 [47]	Increasing neovascularization in the flap	VEGF-165	Plasmid	The flap was constructed 30 days after abdominoplasty. During abdominoplasty, a flap was inserted into fascia. Treatment and electroporation were performed soon after the procedure. In all groups, TRAM was used.	(A) TRAM only group (B) Abdo and PBS-treated group (C) Abdo and empty plasmid-treated group (D) Abdo and VEGF plasmid-treated group	Flap viability and neovascularization	35 days	Transverse rectus abdominis musculocutaneous flap	32 rats	Flap survival was increased in group (D) when compared to groups (A/C). Mean number of vessels per field was increased in group (D) when compared to group (B/C).
Liu et al., 2004 [48]	Increasing flap survival and angiogenesis around flap	VEGF-165	Plasmid	Intradermal injections with a desired substance were performed seven days prior to flap elevation. (A/C) plasmids were within the lipofectamine complex.	(A) VEGF-165 plasmid-treated group (B) Saline-treated group (C) LacZ plasmid-treated group	Flap survival and neovascularization	7 days	Random pattern McFarlane flap	32 rats	The percentage of flap survival augmentation decreased in groups (A/C) when compared to group (B). Blood vessel count was increased in group (A) when compared to group (C).
Liu et al., 2005 [51]	Enhancing survival of ischaemic skin flaps	VEGF-165 PDFG-B bFGF	Plasmid	Substance administered as intradermal injection, seven days prior to flap elevation.	(A) VEGF-165 plasmid-treated group (B) bFGF plasmid-treated group (C) Combined VEGF-165 and b-FGF plasmid-treated group (D) Combined VEGF-165 and PDGF-B plasmid-treated group (E) Triple combined VEGF-165, b-FGF and PDGF-B plasmid-treated group (D) Empty plasmid as control group	Transfection efficiency, neovascularization and flap survival	14 days	Random pattern McFarlane flap	60 rats	Flap survival increased in group (A) when compared to group (F) and in group (C) when compared to group (A). In all groups, there was a significant difference in the number of blood vessels; however, group (E) had the largest number, with group (F) obtaining the smallest number.
Holzbach et al., 2010 [53]	Increasing survival and perfusion of ischaemic skin flaps	VEGF-165	Plasmid	Injections were performed subcutaneously using magnetification and ultrasound, seven days prior to flap elevation. Injection sites placed centrally in the distal half with a distance of one centimetre between one another.	(A1) VEGF- bubbles, magnet and ultrasound-treated group (A2) VEGF-bubbles and magnet-treated group (A3) VEGF-bubbles and ultrasound-treated group (A4) GFP-bubbles, magnet and ultrasound-treated group (P1) VEGF-bubbles, magnet and ultrasound-treated group (D1) VEGF-bubbles, magnet and ultrasound-treated group (A5/P2/D2) NaCl-treated group	Flap survival and necrosis, protein synthesis and micro vessel density	7 days	Rectangular skin flap with a cranial pedicle	46 rats	Reduction in flap necrosis in group (A1) when compared to all the other (A) groups. Flap perfusion index was increased in group (A1) when compared to other (A) groups. VEGF expression was increased in both flap areas in group (P1) compared to group (P2).
Hijjawi et al., 2004 [55]	Increasing flap perfusion and overall flap survival.	PDGF-B FGF-2	Plasmid	Injections were performed subcutaneously. While withdrawing the needle, a plasmid DNA-matrix mixture was evenly injected. This was repeated for the four corners of each section, resulting in the final delivery of the investigated amount.	(A) 2% collagen-treated group (B) FGF-2 plasmid-treated group (C) PDGF-B plasmid-treated group	Flap survival, vascularization and level of cellularity of flaps	7 days	Transverse rectus abdominis muscle flap	24 rats	Flap survival was increased in group (C) when compared to groups (A/B). Flap survival with different PDGF-B plasmid was increased at all concentrations of group (B) when compared to group (A). Vascularity was increased in group (C) when compared to groups (A/B).
Nakagawa et al., 2007 [58]	Increasing angiogenesis and vasodilation of local micro vessels	HGF PGIS	Plasmid	In Exp 1: (A/C) groups had eight injection sites marked over the flap to administer intracutaneously transfected plasmid. (B/D) groups had four injection sites marked in the distal half of the flap. After three days, the flap was elevated, with the remaining pedicle attached at the anterior end and sutured back into place with interrupted sutures. Exp. 1 used Sprague-Dawley rats, while Exp. 2 used GK/JcI rats.	Exp. 1: (A/B) HGF and PGIS plasmid-treated group (C/D) CMV plasmid-treated group Exp. 2: (E) HGF plasmid-treated group (F) PGIS plasmid-treated group (G) HGF and PGIS plasmid-treated group (H) Control group	Concentration of HGF and PGIS, survival rate of flaps and blood flow	7 days	Cranial, pedicled, random-pattern McFarlane musculocutaneous flap	80 rats	Survival rate increased in group (B) when compared to group (D) and in group (G) when compared to group (H). Relative blood flow at 7 days post-op increased in group (G) when compared to group (H).
Fujihara et al., 2005 [71]	Increasing angiogenesis in ischaemic flaps	bEGF	Plasmid	Three intramuscular injection sites, each 100 ug of plasmid DNA in 250 μL PBS. Electroporated afterwards, 2 days prior to flap elevation. In groups (A/B), electroporation was used; groups (C/D) did not use electroporation.	(A/C) Lac.Z plasmid-treated groups (B/D) bFGF plasmid-treated groups	Flap survival and vascularisation (postmortem)	7 days	Dorsal island skin flap	52 rats	Flap survival and vascularization in the distal part of the skin flap was significantly increased in (D) compared with (A/B/C).
Ferraro et al., 2009 [72]	Increasing flap angiogenesis and determining correct concentration	VEGF-165	Plasmid	50 μL intradermal injections of plasmid DNA (2 mg/mL) in sterile saline two days after flap elevation.	(A) VEGF and electroporation-treated group (B) PlasmidVEGF without electroporation-treated group (C) Bare plasmid and electroporation-treated group (D) Intact control group (E) Rats used to determine if delivery of plasmid results in VEGF increase group (F) Rats used to determine kinetics of VEGF and eNOS expression in random skin flap group	Skin survival of the distal 5 cm^2^ of the flap and flap perfusion	14 days	Rostral-based single pedicle random skin flap	51 rats	Flap survival was the greatest in group (A) when compared with other groups. Skin from (A) appeared healthy while skin from the (B) showed evidence of acute inflammation, necrosis and myonecrosis on day 14.
Chang et al., 2021 [74]	Reducing ischaemic necrosis after flap elevation	HIF-1α	Plasmid	Flaps were injected intradermally with 50 µL (1 µg/µL) of plasmid at six designated spots. Injection was administered seven days prior to flap elevation.	(A) CA5-HIF plasmid-treated group (B) Sham plasmid-treated group	Flap survival and gene expression	14 days	Modified island flap based on McFarlane	20 rats	Flap survival increased in group (A) when compared to (B). Mean area of necrosis was smaller* in group (A) when compared with group (B).
Basu et al., 2014 [75]	Accelerating flap healing and decreasing necrosis	VEGF-165	Plasmid	Intradermal injections of respective solutions were administered at the time of flap elevation at different injection sites—two sites (A1/A2/D1/D2) and four sites (B1/B2/C1/C2) Then, the levels of pVEGF in the respective groups were assessed.	(A1) Plasmid-treated group (A2) plasmid and electrode-treated group (B1) Plasmid-treated group (B2) Plasmid and electrode-treated group (C1) Plasmid-treated group (C2) Plasmid and electrode-treated group (D1) Plasmid-treated group (D2) Plasmid and electrode-treated group (E) Control group	Flap survival and flap perfusion	14 days	Standard random dorsal skin flap model (modified McFarlane flap)	109 rats	Flap survival was increased in groups (C2/D2) when compared with control (E). Flap perfusion was increased* in all treated groups (A1/A2/B1/B2/C1/C2/D1/D2) when compared with control (E).
Taub et al., 1998 [77]	Increasing ischaemic flap survival	VEGF-121	Plasmid	Intraarterial injection of the respective solutions was administered via a 30 g needle.	(A) Saline-treated group (B) Control plasmid and lipofectamine-treated group (C) VEGF plasmid and lipofectamine-treated group	Flap survival, dye fluorescence uptake after ligation and angiogenesis	7 days	McFarlane flap	35 rats	Greatest increase in flap survival and vessel lumen diameter was observed in group (A) when compared with other groups (B/C). Greatest* fluorescent dye uptake after ligation was observed in group (C) when compared with other groups (A/B).
O’Toole et al., 2002 [80]	Increasing skin flap survival, measure difference between three VEGF types	VEGF-165 VEGF-167 VEGF-186	Plasmid	Ten equally spaced sites were injected subcutaneously along the length of the midline of the flap (1 mL total).	(A) pVEGF-165-treated group (B) pVEGF-167-treated group (C) pVEGF-186-treated group (D) pEGFP-treated group (E) Saline-treated group	Flap survival, angiography (PM), iron oxide and gelatin, blood vessel count (HE)	7 days	Abdominal skin flap, axially based on epigastric vessels, one pedicle was ligated to render ischaemia	60 rats	Flap survival was the greatest in groups (A/B) when compared to other groups. Angiography showed no significant differences. Histological examination showed considerable variation in the number of blood vessels per slide from individual flaps.
Yang et al., 2005 [81]	Increasing skin flap survival	VEGF-121	Plasmid	Intramuscular injection was directly administered into the panniculus carnosus of the flap at two sites per flap.	(A) p- hVEGF12-treated group (B) pcD2(empty)-treated group (C) Saline-treated group	Flap survival, gene expression, protein expression, vascular density and RBC count in the flap (SPECT)	7 days	McFarlane flap	30 rats	Flap survival was the greatest in group (A) when compared with other groups (B/C). RBC count was the greatest in group (A) when compared with other groups (B/C). VEGF expression was the greatest* in group (A) when compared with other groups (B/C). Vessel number was the greatest in group (A) when compared with other groups (B/C). Vessel density was the greatest in group (A) when compared with other groups (B/C).
Zhang et al., 2005 [83]	Increasing flap survival	VEGF-165	Plasmid	Part I: A 1 ml injection administered subcutaneously with a 25G needle. Punch biopsies performed four days post-operatively. Part II: Injection performed four days prior to flap elevation. Results were obtained five days after flap elevation.	Part I (A) p-VEGF-treated group (B) p-Vax-treated group(C) Saline-treated group Part II (AA) p-VEGF-treated group (BB) p-Vax-treated group (CC) Saline-treated group	Flap survival and IHC/ELISA of punch biopsies (4 days)	Part I: 4 days Part II: 5 days	TRAM flap	52 rats	Flap survival was the greatest in the (AA) group when compared with other groups (BB/CC). Neovascularization increased in (A) when compared with other groups (B/C).
Wang et al., 2006 [84]	Analysis of gene expression and signalling pathway activation in ischaemic flaps	PDGF-B	Plasmid	A 2 mL solution injected intradermally, 7 cm distal from the flap.	(A) pCMVβ-PDGF-B-treated group (B) Saline-treated group	Flap survival and gene expression	7 days	McFarlane flap	20 rats	Flap survival was greatest in group (A) when compared with (B). When assessing gene expression, group (A) showed higher levels when compared with (B). Group (A) showed increased NF-kB expression when compared with (B).
Rezende et al., 2010 [88]	Increasing ischaemic flap survival	VEGF-165	Plasmid	Just prior to flap elevation, intradermal injection in chosen area (numbered according to the distance from the pedicle-—area 1 was over the right deep caudal epigastric artery, area 2 was medial to area 1, area 3 was lateral to area 1 and area 4 was distant to area 1) and electroporation with three pulses of 80 V at the centre of the chosen area, 50 milliseconds with 1 s interval between the pulses.	(A) plgT.VEGF165—area 4 group (B) plgT.VEGF165—area 2 group (C) pl-gT area 4 group (D) pl-gT area 2 group (E) pSVLacZ—area ½ group (F) no plasmid—none group	Flap survival	5 days	TRAM (unipedicle) with four areas determined with increasing distance from pedicle	49 rats	Flap survival was the greatest in group (B) when compared with other groups. Strong necrosis in all groups except (B); preservation of muscular layer* only in (B).
Jafari et al., 2017 [93]	Increasing ischaemic flap survival	HGF	Plasmid	Four sites of intradermal injections (25 μL each), three located in the midline within flap and one outside; eight pulses of 200 V/cm (for 10 ms) using a pulse generator (BTX Gemini X2 System).	(A) HGF, 24 h pre-op-treated group (B) HGF, 24 h post-op-treated group (C) No treatment group	Flap necrosis (planimetry), Doppler and IHC (qHGF)	7 days	McFarlane flap	15 rats	Flap survival was the greatest* in groups (A/B) when compared to control (C). Laser index was the greatest* in group (A) when compared with other groups (B/C). In a semiquantitative histological assessment, inflammatory cell score was lowest* for (A) when compared with other groups (B/C). CD31+ and vessel density was the highest* for (A/B) when compared with control (C). HGF IHS optical density was highest* for (A/B) when compared with control (C).
Jafari et al., 2021 [94]	Increasing ischaemic flap survival	IL-10 HGF VEGF-165	Plasmid	One midline longitudinal injection 1.5 cm away from the edge of the flap (100 μL); subsequent electroporation at eight pulses of 200 V/cm, for 10 ms, using a stainless tweezer rode electrode.	(A) pl-IL-10 24 h pre-OP and pl-hVEGF, 24 h post-OP-treated group (B) pl-IL-10 24 h pre-OP and pl-hHGF 24 h post-OP-treated group (C) No treatment, normal flap elevation group	Flap necrosis (planimetry), histology, IHC for expression of target proteins, fluorescence and lifetime imaging of NADH	7 days	McFarlane flap	15 rats	Flap survival was greatest for (A) than in (B/C). Mean vessel density was higher in (A/B) than in control (C).
Chang et al., 2019 [95]	Increasing flap survival	HIF-1α	Plasmid	Seven days before flap elevation, predetermined flap areas were intradermally injected with viral/saline mixture in six points. During flap elevation, underneath the raised fascia, a sterile silicon sheet was implanted between the flap and the underlying muscle layer. Then, flaps were sutured back to the original position.	(A) NTC9385 CA5-HIF-1α plasmid-treated group (B) Saline-treated group	Efficacy of transfection, area of necrosis, percent of rats with beneficial treatment, histological analysis of pedicle skin flap tissue and CD31-positive blood vessels per HPF	14 days	Modified McFarlane flap model	20 rats	At 24- and 48-h post-op, mRNA expression of HIF-1α was increased in group (A) in comparison to group (B). At days 1 and 7 post-op, area of necrosis was decreased in group (A) compared to group (B). The percentage of rats with beneficial treatment was increased at days 1, 7 and 14 in group (A) compared to group (B). Over the course of treatment, group (A) had lower histological scores compared to group (B). Number of CD31-positive blood vessels per HPF at day 14 was increased in group (A) when compared to group (B).
Lasso et al., 2007 [49]	Increasing overall survival rate of flaps and angiogenesis	VEGF-A 165	Cells (endothelial cells)	An endothelised fibrin scaffold was placed on the cartilage stripped of perichondrium, and the flap was sutured over the scaffold. In (A/C), the pedicle was divided after five days post-op, while in (B/D), the pedicle was divided after two days post-op.	(A/B) Endothelial cells in scaffold-treated group (C/D) VEGF-secreting endothelial cells in scaffold-treated group	Mean flap survival and capillary density	6 days	Axial flap from dorsal region of the ear	32 rabbits	Flap survival, mean number of CD31-positive microvessels and mean number of VEGF-positive vessels increased in group (C) when compared to group (A), and in group (D) when compared to group (B).
Machens et al., 2002 [56]	Induction of therapeutic angiogenesis in ischaemic flaps	PDGF-AA	Cells (fibroblasts)	Intramuscular injections were administered into the panniculus carnosus at evenly distributed sites. Each flap remained connected to the right inferior epigastric pedicle. All (-1) groups’ procedures were performed seven days prior to flap elevation and (-2) groups procedures were performed during flap elevation.	(A1/A2) GMFB and medium-treated group (B1/B2) NMFB and medium-treated group (C1/C2) medium-treated group (D1/D2) NaCl-treated group	Flap necrosis and angiogenesis	7 days	Epigastric island flap	80 rats	Increase in flap survival in group (A2) when compared to all the other groups (A1/B1–2/C1–2/D1–2). Number of capillaries was increased in group (A2) when compared to all the other groups (A1/B1–2/C1–2/D1–2).
Machens et al., 1998 [57]	Increasing angiogenesis in flaps	PDGF-AA	Cells (fibroblasts)	In all experiments, the right-sided flap was subjected to experimental treatment, whereas the left-sided flap served as control. Treatment during flap elevation included evenly distributed intramuscular injections into panniculus carnosus.	(A) PDGF producing fibroblasts and medium-treated group (B) Non-modified fibroblasts and medium-treated group (C) Medium-treated group	Flap survival and protein secretion	7 days	Epigastric island flap	180 rats	At day 2, 3 and 4 post-op, flap survival was increased in group (A) when compared to groups (B/C). Histologically, fibroblasts persisted in all flaps of (A) and (B) without major secondary inflammation of GVHD.
Leng et al., 2017 [63]	Increasing overall flap survival and angiogenesis	F-5 gene fragment of Hsp90-α	Cells (umbilical cord MSCs)	Injections made subdermally at ten points over the flap with blood supply occluded for six hours	(A) hUC-MSCs Ad-“F-5”-treated group (B) hUC-MSCs Ad-null-treated group (C) Pure hUC-MSCs-treated group (D) No injection	Average necrotic area	7 days	Abdominal perforator skin flaps	96 rats	Flap survival was increased in group (A) when compared with other groups (B/C/D) and in groups (B/C) when compared to group (D). In group (A), capillaries were large and uniformly distributed. In group (B/C), capillary density was lower*. In group (D), small capillaries were observed.
Chen et al., 2011 [73]	Improving the survival rate for ischaemic skin	VEGF-165	Cells (myoblasts)	Subdermal injections were administered to three 1 cm^2^ areas (3, 5, and 7 cm from pedicle) to each flap. Two flaps in each rat: The left acted as the control (A2/B2/C2/D2) and received encapsulated non-transfected cells, and the right acted as the treatment group (A1/B1/C1/D1) and received transfected cells. They were assessed at four different timings: at time of elevation (A1/A2) and 2 (B1/B2), 4 (C1/C2) and 7 (D1/D2) days prior to flap elevation.	(A1/B1/C1/D1) VEGF and cells-treated group (A2/B2/C2/D2) control/not transfected cells-treated group	Flap viability, VEGF concentration and neovascularisation	7 days	Rectangular full-thickness flap	64 rats	Both flap survival and vascular counts were increased in treated groups (A1/B1/C1/D1) when compared with control groups (A2/B2/C2/D2).
Rinsch et al., 2001 [79]	Increasing skin flap survival	VEGF-121 VEGF-165 FGF-2	Cells (myoblasts)	A 1 cm long capsule was fabricated using capsules of microporous polyether sulfone secreting either VEGF or FGF-2 and was positioned longitudinally on the subcutaneous tissue at the distal end of the skin flap.	(A) No capsule-treated group (B) C2C12-treated group (C) C2C12 and vector-treated group (D) C2C12 and VEGF121-treated group (E) C2C12 and VEGF165-treated group (F) C2C12 and FGF-2-treated group	Flap necrosis, microangiography and angiogenesis	7 days	McFarlane flap	86 rats	Flap survival and perfusion were greatest in group (F) when compared with other groups.
Yi et al., 2006 [85]	Increasing flap survival	VEGF-165	Cells (endothelial progenitor cells)	Subcutaneous injections of 0.5 mL three days prior to flap elevation.	(A) VEGF-hEPC-treated group (B) hEPC-treated group (C) Culture medium treated group	In vitro: MTT assay. In vivo: Flap survival, plasma VEGF levels and perfusion	28 days	Cranially based flap	30 mice	Flap survival was greatest in group (A) when compared with other groups (B/C). MTT assay showed highest levels for group (A) when compared with other groups (B/C). Serum VEGF levels were highest in group (A) when compared with other groups (B/C). Flap perfusion was the greatest in group (A) when compared with other groups (B/C).
Zheng et al., 2008 [86]	Increasing flap survival	VEGF-165	Cells (MSCs)	A 1 mL subcutaneous injection of solution was administered to the respective groups. Flap elevation was performed four days after injection.	(A) VEGF and MSCs-treated group (B) MSCs-treated group (C) Only medium-treated group	In vitro: VEGF expression. In vivo: Flap survival, perfusion and histologic assessment, plasma VEGF	14 days	McFarlane flap	30 rats	Lap survival was greatest in group (A) when compared with other groups (B/C). Highest perfusion ratio was observed in group (A) when compared with other groups (B/C). VEGF plasma levels were, at all timepoints, significantly higher* in (A) than in (B/C).
Spanholtz et al., 2009 [87]	Increasing flap survival	VEGF-165	Cells (fibroblasts)	Different elevation times after injections—14 days prior to flap elevation (A1/B1/C1/D1), 7 days prior (A2/B2/C2/D2) to flap elevation and intraoperatively (A3/B3/C3/D3). Ten or twenty locations served as injection sites: 10 locations within the flap and 10 locations in the surrounding wound margin—A (flap alone) and B (flap and surrounding), and therefore: A1A or A1B. Each injection delivered 5 × 10^5^ cells in 0.05 mL.	(A1/A2/A3) VEGF-FB-treated groups (B1/B2/B3) AdZ.GFP-FB-treated groups (C1/C2/C3) FB, nonmodified-treated groups (D1/D2/D3) only medium treated groups	In vitro: VEGF expression. In vivo: Flap survival and blood vessels quantity (histology and anti-CD31 IHC)	7 days	McFarlane flap	80 rats	Flap survival was the greatest in group (A2B) when compared with other groups. Blood vessel count was the highest in group (A2B) when compared with other groups.
Spanholtz et al., 2011 [89]	Increasing ischaemic/non-ischaemic flap survival	VEGF-165 bFGF	Cells (fibroblasts)	Total of 40 injection sites: 20 within flap (A—flap alone), 20 in the flap surrounding (B—both), subdermal injections, 1 or 2 weeks before flap elevation. Results were assessed at different times: 14 days prior (A1A/A1B/B1A/B1B/C1A/C1B/D1A/D1B/E1A/E1B) and 7 days prior (A2A/A2B/B2A/B2B/C2A/C2B/D2A/D2B/E2A/E2B) to flap elevation. (All groups marked A were injected into the flap alone, e.g., E1A, whereas with B, injected into flap and surrounding, e.g., D1B).	(A1A/A2A/A1B/A2B) bFGF and VEGF FB-treated groups (B1A/B2A/B1B/B2B) bFGF FB-treated groups (C1A/C2A/C1B/C2B) non-modified FB-treated groups (D1A/D2A/D1B/D2B) pAdcos45. GFP FB-treated groups (E1A/E2A/E1B/E2B) DMEM-treated groups	Flap necrosis and histology	14 days	McFarlane flap	320 rats	Flap survival was greatest in groups (A2B/A2A/B1B) when compared with other groups. Blood vessel density was the greatest in groups (A2B/A2A) when compared with other groups. Statistically significantly higher* number of arterial vessels in groups (A2B/A2A/A1B) when compared with other groups.
Zhang et al., 2011 [96]	Increasing ischaemic flap survival	SDF-1α	Cells (MSCs)	After 24 h of transduction, all cells and Ad-SDF-1α were prepared in 0.5 mL normal saline. The prepared normal saline was injected intravascularly proximal to the femoral artery before wound closure. Pedicle ligation was performed on the fifth day post-op.	(A) Ad-SDF-1α-transduced MSCs-treated group (B) MSCs-treated group (C) Ad-SDF 1α-treated group (D)Normal saline group	Flap survival, MSCs survival, SDF-1α protein expression and microvessel density	10 days	Epigastric free flap	24 rats	Flap survival was the greatest in (A) when compared with other groups. Flap survival areas in (B/C) were greater than those in (D). Expression of SDF-1α was significantly higher in groups (A/B/C) than in (D); overall highest in (A). Mean vessel density was the highest in (A) when compared to other groups. Mean vessel density was higher* in (B/C) when compared with (D) but lower than in (A).
Luo et al., 2021 [97]	Increasing flap survival	SDF-1α	Cells (fibroblasts)	After shaving the hair, an area of 3 × 9 cm was marked on the flap area under aseptic conditions. Then, the full thickness skin flap was elevated, fascia was removed and axial blood vessels entering the flap from the pedicle were excised. All the groups were injected with their respective suspensions at a cell density of 2 × 10^6^/cm^2^. The 0.1 mL solutions were injected intradermally into a total of 18 injection points.	(A) PBS-treated group (B) Luciferase modRNA-treated group (C) SDF-1α modRNA-treated group	Flap survival histology: HE staining, IHC, immunofluorescence, blood flow assessment, tissue oedema and gene expression	10 days	Random flap model in rat dorsum	60 rats	Flap survival was the greatest for (C) when compared with other groups. Flap neovascularization was the greatest in (C) when compared with other groups.

VEGF—vascular endothelial growth factor, FGF—Fibroblast growth factor, Ad—adenovirus, GFP—green fluorescent protein, PDGF—platelet-derived growth factor, TRAM—transverse rectus abdominis musculocutaneous flap, abdo—abdominoplasty, LacZ—beta galactosidase, post-op—post-operatively, ANG—angiopoietin, PBS—phosphate buffer saline, bFGF—basic fibroblast growth factor, eNOS—endothelial nitric oxide synthase, GMFB—genetically modified fibroblasts, NMFB—non-modified fibroblasts, GVHD—graft versus host disease, HGF—hepatocyte growth factor, PGIS—prostacyclin synthase, CMV—cytomegalovirus, TGF-β—transforming growth factor beta, ESW—electrocorporal shock waves, AAV—adeno-associated virus, HSP 90 α—heat shock protein 90 alpha, hUC-MSCs—human mesenchymal stem cells isolated from umbilical cord, RLX—relaxin, DKK—Dickkopf, PLA2G4E—gene encoding phospholipase A2 group IVE, NDI—N-dodeclimidazole, shRNA—short hairpin RNA, pfU—plague-forming units, bEGF—basic epidermal growth factor, p—plasmid, HIF-1α—hypoxia inducible factor-1α, KGF—keratinocyte growth factor, magnet—magnetification, HPF—high-power field, SDF-1α—stromal cell derived factor 1α, MSCs—mesenchymal stem cells, EGFP—enhanced green fluorescent protein, DMEM—Dulbecco’s modified eagle medium, NOS—nitric oxide synthase, ODN—oligodeoxynucleotide, IHC—immunohistochemistry.

**Table 4 ijms-25-10330-t004:** Overview of studies that focused on optimization.

1st Author; Year	Target Gene/s	Vector	Surgical Technique	Experimental Groups	Outcome/s	Observation Period	Flap Model	Number of Animals and Species	Final Results of Therapy (* = Insignificant)
Michaels et al., 2006 [6]	β-gal	Virus	Arterial perfusion using pump was performed in all groups except control. Control groups were injected with viral bolus into veins and intramuscularly.	(A) Ad-LacZ administered through arterial pedicle at 120 mmHg (B) Perfusion with dwell times of 30, 90 and 150 min at 25 °C (C) Perfusion with different dwell temperatures (4 °C, 25 °C and 37 °C) (D) Control group	Gene expression and protein production	35 days	Superficial epigastric free flap Free quadriceps muscle flap	23 rats	The systemic injection group demonstrated a broad, low-level β-galactosidase activity in all harvested tissues, whereas in the ex vivo group, β-galactosidase activity was 20-fold higher in the transduced flap than in any other tissue. ELISA results showing similar* activity levels in the intravascular and intramuscular flap perfusion groups.
Agrawal et al., 2009 [106]	Luc	Plasmid	Delivery of solutions was performed via different routes (drop-wise onto the deep surface of the flap, injected into the flap, intraarterial, intravascular with microbubbles). Plasmid DNA was administered to all groups, but for adenoviral group, only the first three methods were performed. Flaps were then transduced ex vivo.	(A) Topical bathing gene delivery group (B) Direct intra-flap injection gene delivery group (C) Intravascular injection gene delivery group (D) Non-treated group	Gene expression and total radiance	28 days	Superficial inferior epigastric flap (adipo-fascio-myocutaneous)	344 rats	Highest expression of luciferase was observed in (A) when compared with other groups.

β-gal—β-galactosidase, Luc—Luciferase.

## Data Availability

Data are contained within the article.

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
