# Peer review of "Biologic Brachytherapy: Genetically Modified Surgical Flap as a Therapeutic Tool—A Systematic Review of Animal Studies"

_ijms, 2024, doi:10.3390/ijms251910330_

Round 1

Reviewer 1 Report

Comments and Suggestions for Authors

The current systematic review gathers preclinical animal models in the use of genetically modified flaps as therapeutic tool. The paper presents a very extensive research, aiming to summarize a great number or articles. Given the interest to research community, good writing and good work by the authors, the paper should be accepted for publication. However, few suggestions are provided to improve readability of the manuscript and to better focus the message:

1) Figure 1 would benefit of including some schematic of the different strategies / injectables used in biologic modification (virus, plasmid, molecule, cells).

2) Section 2.1: Was the selection criteria solely based in the fact that it included flaps in an animal model? Was there a more extensive selection?

3) Tables are quite extended and could be difficult to follow, specially if looking for a specific strategy , molecule etc. This reviewer thinks that tables would benefit from sorting studies logically. Column sorting (first sorted all the Virus strategies, then all the plasmids, etc.).  would improve readability. 

4) Whereas text is a summary of what is presented in the table, a brief summary offered by the authors on preferred models and overall results in function of the technique, would be beneficial for the manuscript.

5) Table 2 should be split in 2 or 3 tables, in function of the biological agent. The studies including cells could also indicate further information on the cell used.

Author Response

We sincerely appreciate your time to review our paper and provide insightful feedback. We have carefully reviewed all the comments and have made every effort to address each one.

Comment 1: Figure 1 would benefit of including some schematic of the different strategies / injectables used in biologic modification (virus, plasmid, molecule, cells).

  • Figure 1 has been modified. We have added different strategies used for biologic modification.

Comment 2: Section 2.1: Was the selection criteria solely based in the fact that it included flaps in an animal model? Was there a more extensive selection?

  • Yes, extensive criteria were used for study selection, however as the study protocol is fully available in PROSPERO database we aimed to shortly describe the criteria in the manuscript. However, as proposed, we’ve made changes to the manuscript. We added an additional Supplementary File and the following lines in the Materials and Methods (lines 62-64): “Here, we only summarized the study protocol, which is available in full as supplementary file (File S1) and in the PROSPERO database using aforementioned registration number”.

Comment 3: Tables are quite extended and could be difficult to follow, especially if looking for a specific strategy , molecule etc. This reviewer thinks that tables would benefit from sorting studies logically. Column sorting (first sorted all the Virus strategies, then all the plasmids, etc.).  would improve readability. 

  • All tables were sorted according to vector type (1. Virus, 2. Plasmid, 3. Cells 4. Etc). We believe that after the sorting tables are easier to follow.

Comment 4: Whereas text is a summary of what is presented in the table, a brief summary offered by the authors on preferred models and overall results in function of the technique, would be beneficial for the manuscript.

  • We have added a paragraph in the Discussion (lines 373-387), which describes which vector strategy (virus, plasmid, cells, other) seems to have most benefits for clinical translation and guiding towards next developments.

Comment 5: Table 2 should be split in 2 or 3 tables, in function of the biological agent. The studies including cells could also indicate further information on the cell used.

  • As a whole Table 2 corresponds to studies which focused on increasing flap survival – all biological agents used in this group aimed to promote flap survival. Therefore, we’re not able to logically divide the studies into 2-3 tables. As proposed, we have included information regarding the cell type used in the vector subsection in brackets for each study in the Table.

Reviewer 2 Report

Comments and Suggestions for Authors

The authors conducted an interesting systematic review on genetically modified flaps. Here, they try to focus on the flaps serving as "biological Brachytherapy".

However, some minor changes should be made before acceptance.

General:

I fully understand, that the authors analyzed 77 studies on genetically modified flaps. However, only 12 studies treat secondary conditions and therefore can serve as the stated "biological brachytherapy". I don to recommend to remove the other studies, but the focus should be more on the actual relevant studies. Therefore I suggest to shorten the paragraphs 1.2cc in favor of expanding the paragraph actually within the scope of the article.

Minors:

l.192-194: I recommend not to speak about the potential of long term treatment for systemic protein deficiency when protein expression was detected last at four weeks.

l.199: the necrosis of transplanted tissue is not called apoptosis 

Author Response

We greatly appreciate your time in reviewing our paper and offering valuable feedback. We have thoroughly considered all your comments and have made every effort to address each of them.

General: I fully understand, that the authors analyzed 77 studies on genetically modified flaps. However, only 12 studies treat secondary conditions and therefore can serve as the stated "biological brachytherapy". I don to recommend to remove the other studies, but the focus should be more on the actual relevant studies. Therefore I suggest to shorten the paragraphs 1.2cc in favor of expanding the paragraph actually within the scope of the article.

  • We have modified the whole Results section, and we have divided the included studies into 4 groups instead of three. We have shortened paragraph 3.3 (Genetically modified surgical flap for flap-surgery related complications management) as it is the least relevant section. Furthermore, we have added a new section “3.2 Genetic flap preconditioning affecting surrounding tissue and the flap” of studies including allotransplantation and irradiation damage management, which traversed flap-focused therapy and lay in between secondary conditions treatment and treatment of flaps as in these studies flaps actively acted as source of additional biologic activity affecting surrounding tissues. Additionally, we have modified Figure 4 to accustom to the recent changes.

Minors:

Comment 1: l.192-194: I recommend not to speak about the potential of long term treatment for systemic protein deficiency when protein expression was detected last at four weeks.

  • We have extended the description with “Thanks to this therapy, a significant systemic expression of coagulation factor IX was detected for much longer – reported for the last time at four weeks with no further follow-ups, which shows that genetically modified surgical flaps may serve not only for local disease treatment but potentially as a long-term therapeutic tool for systemic protein deficiency disorders.

Comment 2: l.199: the necrosis of transplanted tissue is not called apoptosis

  • We have removed “and apoptosis” from the sentence.